# Engineered Extracellular Vesicles: Emerging Therapeutic Strategies for Translational Applications

**DOI:** 10.3390/ijms242015206

**Published:** 2023-10-15

**Authors:** Jessica N. Ziegler, Changhai Tian

**Affiliations:** Department of Toxicology and Cancer Biology, University of Kentucky, Lexington, KY 40536, USA; jnzi222@uky.edu

**Keywords:** engineered extracellular vesicles, therapeutic applications, heart failure, neurological disorder, cancer

## Abstract

Extracellular vesicles (EVs) are small, membrane-bound vesicles used by cells to deliver biological cargo such as proteins, mRNA, and other biomolecules from one cell to another, thus inducing a specific response in the target cell and are a powerful method of cell to cell and organ to organ communication, especially during the pathogenesis of human disease. Thus, EVs may be utilized as prognostic and diagnostic biomarkers, but they also hold therapeutic potential just as mesenchymal stem cells have been used in therapeutics. However, unmodified EVs exhibit poor targeting efficacy, leading to the necessity of engineered EVS. To highlight the advantages and therapeutic promises of engineered EVs, in this review, we summarized the research progress on engineered EVs in the past ten years, especially in the past five years, and highlighted their potential applications in therapeutic development for human diseases. Compared to the existing stem cell-derived EV-based therapeutic strategies, engineered EVs show greater promise in clinical applications: First, engineered EVs mediate good targeting efficacy by exhibiting a targeting peptide that allows them to specifically target a specific organ or even cell type, thus avoiding accumulation in undesired locations and increasing the potency of the treatment. Second, engineered EVs can be artificially pre-loaded with any necessary biomolecular cargo or even therapeutic drugs to treat a variety of human diseases such as cancers, neurological diseases, and cardiovascular ailments. Further research is necessary to improve logistical challenges in large-scale engineered EV manufacturing, but current developments in engineered EVs prove promising to greatly improve therapeutic treatment for traditionally difficult to treat diseases.

## 1. Introduction

Since their discovery in the 1980s, researchers have held extracellular vesicles (EVs) under great scrutiny as potential therapeutic agents for intractable diseases. EVs are small spheres released by almost every mammalian cell type and are placed into one of three categories based on their biogenesis and have slight variations in size. The smallest EVs, exosomes, are only 30–100 nm wide. Exosomes formed as mature endosomes fold inward to create a multivesicular body (MVB) which then fuses with the cellular membrane, releasing their cargo of numerous exosomes [1,2]. Microvesicles come in a sundry of sizes, from 50–1000 nm [3]. They are released directly from the membrane of the parent cell through the cellular membrane bubbling outward to produce the vesicle [2,4]. Finally, apoptotic bodies vary wildly from 100–5000 nm and are formed, as the name suggests, when a cell undergoes apoptosis [4]. Apoptotic bodies are not currently an area of extensive research, especially compared to the amount of research that is currently being and has recently been performed on exosomes. In vivo, EVs are primarily used as methods of communication between cells and to change the function of the cells that receive them through their cargo [3]. Thus, they are known to carry a wide variety of mRNA, ncRNA, proteins, DNA, and lipids [3,5]. To deliver their cargo to the target cell, they must be able to fuse and bind with it. Thus, their membrane contains many proteins to facilitate their binding and fusion to the target cell and proteins reminiscent of the parent cell of the EV. For example, those derived from dendritic cells oftentimes contain both MHC I and II proteins [5,6].

The fluid that contains EVs is easy to collect as almost every bodily fluid, including urine, breastmilk, blood, and tears, contains EVs [1,2,4]. However, the purification of EVs poses a more difficult challenge. The current standard for the purification of EVs from a fluid sample is ultracentrifugation (UC). However, UC comes with a high price tag, and the process may incur damage to the EVs, leading to low yields of EVs [1,4]. Other methods of purifying EVs have been tested such as size exclusion chromatography, but this may lead to the accidental inclusion of lipoproteins and other similarly sized particles also found in bodily fluids [4]. Immunoaffinity capture is another method that allows for the targeting of specific types of EVs from a sample but is overwhelmingly expensive and, again, only provides low yields of EVs. Most recently, asymmetric-flow field-flow fractionation was developed [1]. This method uses a countercurrent flow—one laminar and one variable cross flow—to separate particles based on their hydrodynamic properties and density [7]. Unfortunately, none of these methods can provide the high yield, reasonable cost purification necessary for large-scale, commercial use of EVs as therapeutics. Thus, more research is required on how to have high accuracy and high-yield purification of EVs without high costs.

In addition to naturally being present in most bodily fluids, EVs naturally contain different types of cargo depending upon their parent cell and the required response of the target cell. For example, EVs released from cancer cells contribute to tumor metastasis. These EVs mediate immunosuppression and the creation of a tumor-friendly microenvironment by containing cargo that inhibits apoptosis in the target cell and instead increases the growth rate or provides drug resistance [8]. EVs also exhibit innate homing capabilities depending upon their parent cell and the peptides attached to their surface. For example, EVs from mesenchymal stem cells (MSC) have the native ability to accumulate and localize in ischemic heart tissue [9]. Another study found that MSC-EVs innately homed to tumorous regions [10]. While naturally derived EVs can pose as potential therapeutic agents unaltered, engineered EVs pose the utmost therapeutic potential [11] because EVs are inherently unlikely to elicit an immune response [3,9,12,13], can pass the blood–brain barrier [10,12,14], and can be modified to carry the therapeutic cargo and target specific cell types.

Therefore, there are two ways to modify the cargo of EVs: endogenously or exogenously. To do so endogenously, one must cause the parental cell to overexpress the desired payload, be it an mRNA, miRNA, protein, etc. Then, through diffusion, the EVs released from that cell will naturally contain more of that as their cargo. To modify the payload of the EVs after their release from the parent cell physical or chemical methods must be employed [4]. These methods include incubating the EVs with the desired cargo, electroporation, sonication, or simply coating the load—usually a drug in this case—in a synthetic EV membrane from neutrophils [11]. Incubation is the least likely to cause damage to the EVs but only exhibits reliable success with small, hydrophobic molecules, leading to the necessity of other, more invasive methods to load molecules that do not meet those specifications [4].

Furthermore, targeting peptides may also be attached to the EV. Targeting peptides serve to increase the uptake of the EVs by a specific cell type, thus decreasing their innate accumulation in the reticuloendothelial system [15,16]. However, the degree of accumulation in these organs is dependent upon the EV injection site and the parental cell of the EV [15]. The targeting peptides that decrease the accumulation in the liver and spleen can be attached in a few ways. Similar to the endogenous method of loading cargo, the parental cells of the EVs could be transfected to express or overexpress a specific protein with the peptide attached, causing it to naturally appear in the membrane of EVs [17]. The addition method known as “click-chemistry” in which strong chemical bonds are made under physiological conditions [18], provides another approach that is not cytotoxic and can be performed after the EVs have been produced, for up to 24 h post isolation [17]. The final method, cloaking, involves a membrane with attached peptides and an anchor point to attach it to the EV itself [19]. The rapid progress in developing methods by which EVs may be engineered to express certain targeting sequencings and contain virtually any biomolecular cargo desired inspires much hope for the future of medicine and the treatment of notoriously difficult-to-treat diseases. Tremendous efforts have been made to develop EVs as next-generation delivery tools for drugs and nuclear acid molecules (i.e. microRNAs) [20] due to their innate ability to transport biomolecules between cells, and it is generally accepted that EV-based drug delivery systems are superior to synthetic nanoparticle system [21]. Additionally, EVs possess an unprecedented advantage for central nervous system applications by their innate ability to cross the blood–brain barrier (BBB) and effective endocytosis pathways resulting in a much higher internalization than other nanoparticles. Importantly, with the development of stem cell technologies, especially induced pluripotent stem cell (iPSC) technology, it becomes possible to produce personalized iPSC-derived EVs at a high efficiency because iPSCs are easy to obtain through the direct reprogramming of patient’s somatic cells and can produce much more EVs than human mesenchymal stem cells [22], which will greatly improve the immunogenicity and avoid potential ethical issues. Therefore, these engineered EVs, especially personalized iPSC-derived EVs, hold the potential to provide extensive therapeutic application in humans diseases, including but not limited to cancers, neurological disorders, and cardiovascular diseases. To highlight these potential therapeutic applications of engineered EVs in human diseases, we summarized the recent research progress in the field.

## 2. Engineered EVs in Cancers 

Because of their unique ability to act as a delivery vehicle in vivo, EVs pose as excellent potential therapeutic agents for cancer treatment; many researchers have looked into this potential, and excitingly, the research is proving promising. EVs derived from cancer cells naturally exhibit a cancer-cell homing ability but oftentimes contain cargo that is beneficial to the tumor. Thus, the natural cargo can be removed via electroporation but the tumor-homing ability is maintained and the desired cargo may be added [23]. Various cancer cell types overexpress the integrins αv and β3/β5 both within the tumor and within their vasculature [24], which makes them useful as a potential target for EVs to hone to, and thus the RGD peptide was created [25]. This peptide is relatively easy to conjugate to the EVs through a simple click-chemistry reaction [26], making it a suitable choice for translational applications both from a therapeutic and logistical standpoint. Membrane cloaking is another method of increasing the targeting efficiency of EVs; that is, the addition of cell-type or tissue-specific targeting units to the membrane of the EV with the addition of an anchor. Although it is not yet widely researched, cloaked EVs exhibit an increased uptake efficiency, and the target cell is much more likely to uptake cloaked EVs [19]. EVs have already shown promising potential as therapeutic delivery vehicles for numerous cancers, several of which will be discussed hereafter. Incredibly, none of the researchers in these studies found any toxicity due to the EVs as the mice treated.

For example, glioblastoma (GBM) is arguably the deadliest of all human cancers, and most professionals currently believe it to be incurable [27]. However, it responded significantly to treatment with engineered EVs. In a study by S. Hong et al. in 2021, they found that by engineering EVs to carry miRNA-124, they were able to lower the expression of an anti-apoptotic marker and a proto-oncogene, *mcl-1* and *c-Myc*, respectively. Thus, the apoptosis rate increased in GBM tumor cells, and the proliferation rate decreased. [13]. However, another study on miRNA-124 found that it prolonged the lifetime of M2 macrophages—the anti-inflammatory variant—and inhibited the M1 phenotype of macrophage—the variant that is aggressively anti-cancer [28]—which highlights the necessary precision of engineered EVs to target a specific cell type so as not to cause adverse effects. Another group of researchers looked into the potential of EVs to target GBM tumor cells using a targeting peptide known as c(RDGyK), a cyclic version of the standard αvβ3 integrin targeting ligand [29], and found it was easy to introduce to the EV [14]. EVs engineered to carry anti-PD-L1 siRNA were delivered to the GBM tumor, with researchers looking to decrease the levels of PD-L1 because this ligand, aptly named the programmed death ligand, allows cancer cells to slip past the immune system and avoid apoptosis [14,30]. When in combination with radiotherapy which increased uptake of EVs by the tumor, EVs successfully increased the number of anti-tumor CD8^+^ T cells and greatly extended the median survival time of mice with GBM tumors [14]. Success with anti-PD-L1 has also been found with the loading of metformin, a drug typically used to treat high blood glucose, into macrophage-derived EVs which then degrades the collagen of the tumor and renders it more likely to infiltration by CD8^+^ T cells [31].

Furthermore, gynecological cancers are of great interest as they lead to the mortality of many women around the globe. For example, because there is no prevention measure for ovarian cancer and screening is not commonplace nor high in accuracy, it is often found once it is in its advanced stages and already causing a host of symptoms for the patient [32]. Thus, many researchers have studied the potential of engineered EVs as therapeutic agents for ovarian cancer. When these EVs were engineered to express the tumor-homing RGD tag, there was an even more profound decrease in angiogenesis of the tumors, thus indicating that RGD helps the receiving cell to uptake the cargo from the tagged EV more efficiently and effectively. These tumor-homing EVs were loaded with miR-92b-3p and successfully suppressed the angiogenesis of the tumors [33]. miR-92b-3p acts as a tumor suppressor by inhibiting the expression of GABRA3, a transporter that is known to have abnormal expression in breast, pancreatic, and lung cancer [34,35]; it may also be abnormally overexpressed in other types of cancers, meaning that miR-92b-3p loaded into EVs could have translational applications for more than just one type of cancer. Another group of researchers looked into the anti-tumor effects of miR-199a-3p loaded into EVs for the treatment of ovarian cancer. They used non-targeted EVs engineered to carry miR-199a-3p, which is traditionally downregulated in several types of cancer compared to non-cancerous cells and is known to cause G1 phase arrest and upregulate c-Met [36,37]. It is also believed to upregulate mTOR. When all of these effects are combined, they produce a powerful anti-cancer effect that inhibits cancer cells and renders them more vulnerable to chemotherapeutic drugs [38]. Additionally, miR-199a-3p-loaded engineered EVs significantly inhibited the growth of ovarian cancer cells, thus decreasing the invasiveness of the tumor [36]. A very recent study suggests that hydrogel-loaded EVs derived from M1 macrophages transfected with Siglec10 are powerful tools for the treatment of ovarian cancer. These EVs successfully suppressed efferocytosis, thus allowing for increased antigen presentation to the now increased levels, due to the EVs, of M1 phenotype macrophages [39]. 

Furthermore, these hydrogel-loaded EVs have also been shown to be effective in treating triple-negative breast cancer (TNBC) that overexpresses CD24 by creating an anti-tumor microenvironment [39]. While breast cancer does have effective regular screening methods, unlike ovarian cancer, the survival rate is still poor, especially for women suffering from TNBC [40]. To treat this type of breast cancer with EVs, engineering them to express Hiltonol-ELANE-α-LA (HELA), in which ELANE is the ICD inducer human neutrophil elastase and α-LA is α-Lactalbumin, a protein expressed in the breast during many breast cancers, was highly effective to increase the targeting of the EVs [41]. To target other breast cancer cells, the epidermal growth factor receptor (EGFR) could be utilized, as this is overexpressed in some breast cancer cell types. However, a challenge arises in that the targeting peptide must target EGFR but not induce mitosis like the innate ligand epidermal growth factor (EGF). There is a ligand that can be used to target EGFR without causing the mitosis effect inherent of EGF. This peptide sequence, YHWYGYTPQNVI, is known as GE11 and was able to successfully be engineered onto HEK293 EVs [42]. EVs may also be fused with liposomes to create a hybrid delivery vehicle and engineered to express anti-EGFR antibodies so that they effectively target breast cancer cells [23]. 

Moreover, when EVs were tagged with HELA and loaded with a breast cancer drug, they effectively inhibited tumor growth in TNBC, even more so than when the drug was delivered freely by itself [41]. Interestingly, in breast cancer cells, the miR-205 is severely under-expressed, leading to increased cancer cell proliferation and greater ease of metastasis for cancer cells as miR-205 regulates cell growth. EVs engineered to contain miR-205 caused a greater number of cancer cells to be in apoptosis and fewer cancer cells to be viable through the decreased expression of an anti-apoptotic gene Bcl-2 [43]. It is probable that if miR-205, which targets the anti-apoptotic gene Bcl-2, was loaded into EVs engineered with the HELA tag, it would be especially effective in treating TNBC cells, but this research has yet to be performed. EVs tagged with the GE11 peptide to target breast cancer cells expressing EGFR were loaded with miRNA let-7a and significantly inhibited the proliferation of the cancer cells through a poorly understood pathway [42]. More research should be performed on the mechanism by which miRNA let-7a inhibits cancer cell proliferation and the effect that it may have when administered in tandem with miR-205 in the treatment of breast cancer. However, as miRNA let-7a downregulates *c-Myc,* a proto-oncogene [13], expression in other cancers [44] leading to chemoresistance and increased tumor aggression [45], it may perform in the same way during breast cancer, leading to the results found by Ohno et al. Doxorubicin (Dox), when loaded into immature dendritic cell-derived EVs tagged with the iRGD peptide was also successful in targeting breast cancer cells and greatly inhibiting tumor growth without toxicity to normal cells [46].

Additionally, as the third leading cause of death in women worldwide, new treatments for cervical cancer are of great interest [47]. When bone marrow mesenchymal stem cell (MSC)-derived EVs were loaded with miR-375, they decreased the proliferation, migration, and invasion capabilities of the cancer cells while simultaneously increasing apoptosis in the cells [48]. In gastric cancer, increased levels of miR-375 lead to ferroptosis [49], which may be the mechanism by which the engineered EVs loaded with miR-375 were able to increase apoptosis in the cervical cancer cells. When tested on mice xenografted with cervical cancer tumors, the engineered EVs decreased the tumor volume and the number of migrated, invaded, and proliferative cells while simultaneously increasing the number of apoptotic cervical cancer cells [48]. If miR-375 promotes apoptosis by inducing ferroptosis in cervical cancer as it does in gastric cancer, translational approaches involving EVs engineered to carry miR-375 may benefit from additional iron, either by supplement or directly loaded into the EVs themselves, or from EVs loaded with Erastin, a ferroptosis inducer [50]. Naturally, more research is required in this area, especially as the authors could not find evidence of metal loaded into EVs, except for the successful delivery of iron oxide through engineered EVs to a mouse brain [51].

Furthermore, malignant melanoma poses a severe threat to all persons, especially those of darker skin tones; for them, it is often not discovered until it is in an advanced stage [52]. There has also been a dramatic increase in the incidence of melanoma, and as it is the cause of more than 85% of skin cancer-related deaths, it is vital to find new therapeutics [53]. MSC-derived EVs engineered to bind to the αvβ3 integrin—an integrin overexpressed in many tumor types—successfully targeted melanoma tumor cells and avoided the liver. Additionally, they were able to slow down the rate of mitosis, promote apoptosis, and suppress invasion of the tumor cells [54]. EVs derived from melanoma cell lines engineered to carry miR-195-5p, which is known to regulate key malignancies of tumor cells [55], were effective in decreasing the proliferation of treated melanoma cells by increasing hypodiploid cells likely through the pathway of one of the 41 miRNAs that experienced changed expression due to treatment with the EVs [56].

Sadly, as the second leading cause of death from cancer around the globe, gastric cancer still poses a considerable threat to public health. EVs were engineered to carry the RNA cargo circDIDO1, which increased the expression of SCOS2 in treated cancer cells, leading to a substantial decrease in cancer cell proliferation rate and development. Many treated cells underwent apoptosis due to the RNA cargo [57], likely because SCOS2, which exhibited increased expression, is known to be a vital aspect of cell growth through the GH/IGH-1 signaling pathway [58]. EVs engineered to deliver circDIDO1 were successful in inhibiting gastric cancer progression via increased SCOS2 expression, and other types of cancers may benefit from these EVs as well because SOCS2 downregulation is common for many types of cancer; however, EVs loaded with this cargo do not pose a therapeutic use towards leukemia as SOCS2 is already overexpressed in those cancer cells [59]. Nevertheless, this study by Guo et al. highlights the powerful effects of RNA-loaded EVs, while at the same time pointing to the immense research that must be conducted to avoid any adverse side effects.

Posing another severe threat to public health, colon cancer has a five-year survival rate of just 52% [60]. EVs were engineered with the target-Her2-LAMP2-GFP (THLG) targeting unit that targets the Her2 protein commonly overexpressed in colon cancer cells. When loaded with miR-21 and 5-FU, a chemotherapeutic drug used to treat colon cancer, EV treatment exhibited a significant increase in the number of tumor cells in apoptosis, and cell proliferation decreased by 82%. Interestingly, the mechanism of this arrest was during the S phase of the cell cycle [61]. Even more intriguingly, miR-21 is overexpressed in ovarian cancer cells. In that cell line, when suppressed, it causes apoptosis of cancer cells while decreasing their proliferation; this is through the PTEN/PI3K/AKT pathway [62]. However, it is also involved in cancer cells gaining resistance to chemotherapeutics [63], likely the mechanism of action found in Liang et al.’s study. Furthermore, when Liang et al. injected mice with colon cancer tumors with these EVs, there was no significant decrease in body weight, but there was a remarkable decrease in tumor weight and inhibition of tumor growth [61].

Additionally, the incidence of pancreatic cancer is increasing, and with over 430,000 pancreatic cancer-related deaths in 2018, the development of new therapeutics is vitally important [64]. Like many other types of tumors, pancreatic tumors often overexpress the αvβ3 integrand targeted by the RGD peptide. Thus, EVs engineered to have the RGD peptide attached and filled with the common pancreatic cancer drug paclitaxel cause significantly decreased tumor growth compared to free-drug delivery. The cells treated with the EVs loaded with paclitaxel were more likely to go into apoptosis than even those treated with free paclitaxel [65]. The homing ability of the EVs with the RGD targeting peptide increased the efficiency and effectiveness of the paclitaxel delivery, indicating that loading chemotherapeutic drugs into EVs could have immense translational applications.

Moreover, liver cancer poses low five-year survival rates at under 50% for all age groups and genders [66]. HepG2 cells, a type of liver cancer cell, overexpress the SR-B1 receptor that interacts with the protein Apo-A1. SR-B1 also exhibits overexpression in many other tumor-associated vascular endothelium. Thus, to target these HepG2 cancer cells using EVs, the researchers added the Apo-A1 protein on the surface of the EV. This targeting protein increased the targeting itself and the ease with which the recipient cell could take up the EV because the binding of the protein triggered receptor-mediated endocytosis of the EV. When researchers loaded these engineered EVs with miR-26a, the target cells exhibited decreased migratory abilities [67]. miR-26a is known to inhibit the proliferation of both thyroid and prostate cancer cells through the targeting of ARPP19 and CDC6, respectively, both of which are genes controlling cell cycle progression [68,69,70,71], thus indicating that miR-26 likely interacted with either of these gene products—or perhaps another, yet unknown cell cycle regulator—to produce the anti-migratory abilities found in the study on liver cancer by Liang et al. [67]. Ferroptosis may also be induced in liver cancer cells when they are treated with EVs loaded with CD47, Erastin—a ferroptosis inducer—and Rose Bengal—a photosensitizer—followed by laser irradiation [50].

Even more devastatingly, osteosarcoma, a type of bone tumor, is especially prevalent in children and adolescents, often requiring chemotherapy and surgery to treat [72]. Long-noncoding RNA Maternally Expressed Gene 3 (lncRNA MEG3), which functions as a sponge for miRNAs [73], is under-expressed in osteosarcoma tissues [74]. Researchers loaded EVs engineered to target the αvβ3 integrin through the engineered expression of the cRGD peptide with lncRNA MEG3, and they successfully targeted osteosarcoma cells in vivo. The engineered EVs induced increased apoptosis of the cancer cells and decreased migration and proliferation. The researchers believed this result was likely due to the sponging effect of lncRNA MEG3 on miR-185-5p [74]. While their results do show inhibited tumor growth, miR-185-5p, when upregulated, prevents the proliferation of leukemia cells while at the same time promoting apoptosis [75], which is the opposite of what is suggested by Huang et al. Thus, more research must be carried out before loading lncRNAs into EVs, as their effects on each type of cancer must be well understood before their therapeutic effects can become widely applied. Nevertheless, lncRNAs have a relatively recently discovered complex role in cancer development and regulate transcription on a cell-specific basis [76]; thus, the ability to directly deliver them to specific cell types using engineered EVs poses great promise for translational applications.

Additionally, engineered EVs are also poised to be an excellent therapeutic tool to sensitize cancer cells to radiotherapy. In one study, EVs derived from M1 phenotype macrophages were not only successful in polarizing both M2 and M0 phenotype macrophages to the anti-tumor phenotype M1 but also were successful in delivering radiotherapy sensitization cargo to tumor cells, thus increasing the level of apoptosis when treated with radiotherapy. Those EVs engineered with an anti-PD-L1 tag were most successful at targeting the tumor cells. Mice with tumors treated with these EVs exhibited significantly decreased tumor weight and a significant increase in survival time. However, the engineered EVs did not induce chronic inflammation or increase the side effects of the radiation [77]. Another study found that EVs engineered with the 131 isotopes of iodine and a targeting peptide were most successful in being cytotoxic to only cancer cells when loaded with the chemotherapeutic agent Dox. They exhibited good biosafety in mice and were well retained within the blood, allowing for a high level of accumulation within the tumor [78]. EVs may also be used to sensitive cancer cells to chemo-sonodynamic therapy through the loading of sonosensitizer agent Chlorin e6 and the addition of a mitochondrial targeting moiety triphenyl phosphonium [79]. The applications of Engineered EVs in developing cancer therapeutic strategies are summarized in Table 1, and the main approaches used for the isolation and characterization of engineered EVs are listed in Table 2.

## 3. Engineered EVs in Cardiovascular Diseases

Amazingly, EVs are useful for more than just cancer treatment. Targeting peptides, such as the sequence CSTSMLKAC, cause EVs engineered to express it to target not only cardiac tissue but specifically infarcted heart tissue [80]. Cardiac tissue likely has further difficulty receiving EVs than other tissue due to the layout of blood vessels in the tight junction cellular arrangement of the heart [81]. During a hypoxemic event, EVs are released by the heart containing different levels of specific mRNAs when compared to those released at normal oxygen levels. Interestingly, they innately target ischemic myocardial cells due to a peptide on their surface [82].

**Table 2 ijms-24-15206-t002:** Main approaches used for the isolation and characterization of engineered EVs.

Type of Diseases	Origin of EVs	Isolation and Characterization	Ref.
GBM	HEK293 T; human neural progenitor cells	**Isolation and purification**: ExoQuick method (EXQ), ultrafiltration (UF), differential ultracentrifugation; **Characterization:** Nanoparticle tracking analysis (NTA), transmission electron microscopy (TEM) and western blot with anti-CD63, CD81, Alix and TSG101 antibodies.	[13,14]
Ovarian Cancer	HEK293 T; fibroblasts; M1 macrophages	**Isolation and purification**: Differential velocity centrifugation, ExoQuick-TC kit; **Characterization:** TEM, TEM with immunogold labeled with anti-CD63 antibody, SEM, NTA and Western blot with anti-CD63, HSP70 antibodies.	[33,36,38,39]
Breast Cancer	Dendritic cells; MDA-MB-231; HEK293 T	**Isolation and purification**: Differential ultracentrifugation; **Characterization:** Flow cytometry; Immunoelectron microscopy and Western blot analysis; electron microscopy and dynamic light scattering	[41,42,43]
Cervical Cancer	Bone marrow-derived MSC	**Isolation and purification**: Differential ultracentrifugation; **Characterization:** TEM, NTA and Western blot analysis	[48]
Melanoma	Melanoma cell lines	**Isolation and purification**: Differential ultracentrifugation; **Characterization:** TEM, NTA and Western blot analysis	[56]
Gastric Cancer	HEK293 T	**Isolation and purification**: Differential ultracentrifugation; **Characterization:** TEM, NTA and Western blot analysis	[57]
Colon Cancer	HEK293 T	**Isolation and purification**: Differential ultracentrifugation; **Characterization:** TEM and dynamic laser scatter (DLS) and Western blot analysis	[61]
Pancreatic Cancer	PANC-1; U937	**Isolation and purification**: EV Concentrate from conditioned medium with Centricon^®^ Plus-70 filters, then subjected to ultracentrifugation; **Characterization:** TEM, NTA and Western blot analysis	[65]
Liver Cancer	HEK293 T	**Isolation and purification:** Differential centrifugation, exoEasy Kit; **Characterization:** TEM, dynamic light scattering (DLS) and Western blot analysis	[50,67]
Osteosarcoma	Osteosarcoma cells	**Isolation and purification:** Total Exosome Isolation reagent (Invitrogen, USA); **Characterization:** TEM, NTA and Western blot analysis	[74]
Alzheimer’s Disease	Dendritic cells	**Isolation and purification:** Differential centrifugation and exosome pulldown assay; **Characterization:** TEM, NTA and Western blot analysis	[12]
Parkinson’s Disease	Dendritic cells	**Isolation and purification:** Serial centrifugation; **Characterization:** None	[83]
Cocaine-induced Neuroinflammation	Dendritic cells	**Isolation and purification:** Differential centrifugation; **Characterization:** NTA and Western blot analysis	[84]
Stroke; Cerebral Ischemia	Brain endothelial cells	**Isolation and purification:** Differential centrifugation; **Characterization:** dynamic light scattering (DLS), a calcein AM flow cytometry assay for membrane integrity of EVs, TEM and Western blot analysis	[85]

Moreover, by adding a cardiomyocyte targeting peptide to an EV containing a siRNA designed to down-regulate a gene involved in the apoptosis of cardiomyocytes, the degree of apoptosis in cardiomyocytes significantly decreased [81]. Furthermore, a study on the effect of EVs engineered to target cardiac cells after myocardial infarction in rats found that they were beneficial in increasing cardiac regeneration, likely due to the miRNA contents of the EVs [86]. Similarly, injected MSCs have been shown to home directly to ischemic heart tissue when engineered to have an additional cardiac homing peptide [9]. However, MSC-EVs have been shown to deliver the same benefits to myocardial cells undergoing repair as when MSCs are delivered into the heart, but do not carry the risk of becoming tumorigenic because they are not able to go through mitosis [87]. Similarly, when bone marrow-derived mesenchymal stem cell-derived EVs are loaded with atorvastatin, a drug used to prevent cardiovascular disease, and are administered to rodents after myocardial infarction, they significantly improve heart function and reduce the size of the infarct, even more so than just MSC-EVs alone [88]. These findings have implications for translational applications because it has been determined through this research that the mode of action for the benefits of transplanted MSCs is the EVs themselves [89], and when tested on healthy human fibroblast cells, it indicates that cardiac targeting peptide will be just as useful in humans as they are in mouse models [16]. However, human clinical trials have yet to be performed. 

## 4. Engineered EVs in Neurological Diseases 

Incredibly, EVs are not just potentially useful for the treatment of myocardial infarctions and other cardiovascular diseases, they have also shown efficacy in treating neurological ailments. The current treatment for Alzheimer’s disease (AD) leaves many patients and their loved ones wishing for more; the current medications can improve the quality of life for the patient but fail to slow down the rate of mental decline or the progression of the disease [90]. In 2011, researchers found that siRNA, loaded into EVs engineered to express the rabies viral glycoprotein (RVG) peptide (YTIWMPENPRPGTPCDIFTNSRGKRASNG), was successfully delivered to the mouse brain through the targeting provided by the RVG peptide. Thus, they conjectured that engineered EVs would be beneficial in treating AD. They successfully loaded BACE1 siRNA—known to interact with the protease BACE1 mRNA—into EVs and targeted them to the brains of mice using the RVG peptide [12]. Because the BACE1 protease is responsible for producing the β-amyloids that are a part of the toxicity of AD, the inhibition of it would slow down the progression of the disease exponentially [12,91].

Additionally, Parkinson’s disease (PD) is a common neurodegenerative disease and is only second to AD oftentimes caused by a combination of risk factors including genetics [92]. Thus, gene expression manipulation through the usage of EVs could mediate this risk factor, reduce the severity and decrease the decline of patients with PD. For example, EVs targeted with the RVG peptide and loaded with shRNA designed to decrease the levels of mRNA that code for alpha-synuclein, a protein highly involved in the pathology of PD, successfully decreased the levels of alpha-synuclein protein in the cells [83].

AD, PD, and Huntington’s disease (HD) all exhibit expression levels of miR-124 that deviate from the norm. This is also seen during continuous illicit drug use, specifically with the use of cocaine, leading to increased neural inflammation due to underexpression of miR-124. EVs derived from dendritic cells were loaded with miR-124 and administered to mice that had also been using cocaine. It was determined that the miR-124-loaded EVs were successful in halting the activation of microglial cells and thus inhibiting the inflammation associated with cocaine usage [84]. This could be used not only for drug-induced neural inflammation, which is vitally important for treating America’s drug crisis, but also in the treatment of AD, PD, or HD.

Furthermore, stroke as a result of cerebral ischemia can quickly become deadly and bring a lifetime of adverse effects on the patient, especially when not treated rapidly because about two million brain cells die each minute [93]. Researchers believed that deriving EVs from a neural progenitor cell line derived from the ventral mesencephalon region of the human fetal brain would likely hold the essential anti-inflammatory characteristics that could mediate the effects of cerebral ischemia. They used click-chemistry to attach the RGD-C1C2 peptide to the EVs and successfully targeted the tissue affected by cerebral ischemia. They found a significant decrease in the levels of proinflammatory cytokines dependent upon the dosage of EVs received [17]. Of course, the use of human fetal tissue always raises a large variety of ethical questions and concerns that must be considered as research continues [94]; nevertheless, the anti-inflammatory benefits of the EVs derived from this cell line should not be pushed aside. As an alternative to human fetal brain tissue-derived EVs, EVs derived from astrocytes may provide similar benefits [95]. Another pathway is to consider the use of brain endothelial cell-derived EVs engineered to contain mitochondria and HSP27, which will serve to decrease the tight junction permeability and protect brain endothelium after a stroke. When tested in mouse models, these EVs induced neuroprotection after stroke [85]. Current therapeutic strategies of engineered EVs have been summarized in Table 3, and the main approaches used for the isolation and characterization of engineered EVs are listed in Table 2.

## 5. Engineered EVs in Other Human Diseases

Incredibly, the potential translational application of engineered EVs expands even further. While the research is still in its infancy, EVs may be loaded with CRIPSR/CasRx (Cas13) to manipulate target RNA instead of permanently changing DNA [96,97]. Researchers successfully loaded EVs with CasRx and a guide RNA set to target mCherry in mCherry transfected cells. The expression of mCherry in the treated cells was greatly diminished compared to the controls. This finding has a profound impact on the therapeutic potential of EVs to treat acute diseases by changing the mRNAs, and thus the proteins, expressed within the cell [97]. An example of an acute infirmity that EVs could eventually treat is COVID-19. While this specific approach has not been taken as a treatment for COVID-19 yet, EVs have been used in the treatment of COVID-19. While these mesenchymal stromal cell-derived EVs were not engineered in any way, COVID-19 patients treated with these nebulized EVs in this pilot study exhibited increased lymphocyte counts and decreased markers of inflammation. Furthermore, no adverse events occurred nor did any worsening of their symptoms. The biosafety of the nebulized EVs was excellent, as no patients experienced any hepatotoxicity or nephrotoxicity [98].

As EVs can easily pass the placenta barrier, EVs engineered to express a peptide that hones to the placenta could deliver cargo that does not adversely affect fetal or maternal health. Surprisingly, some tumor-homing peptides simultaneously target the placenta [99]. More research must be completed on this, but the prospects of this development would be impactful on the gynecological field as many drugs to treat a fetus require either the mother to take the medication or direct administration to the fetus through intracordal, intramuscular, or intraperitoneal injection [100]. The development of a drug delivery method that could target the fetus without causing side effects in the mother would be beneficial. 

Furthermore, while the percentage of deaths due to sepsis in those hospitalized for it has dropped to just under 18% in the United States [101], the global mortality rate for young children and infants with sepsis is still high at 32% [102]. Sepsis often induces acute lung injury/acute respiratory distress syndrome (ALI/ARDS), increasing the mortality rate to a saddening 40% [103,104], and can also induce acute liver injury due to the extreme levels of inflammatory cytokines [89]. Thus, researchers engineered EVs derived from mesenchymal stem cells to carry MiRNA miR-26a-5p, which is known to be underexpressed in mice experiencing sepsis, and successfully delivered the cargo via EVs to the liver cells [89]. This downregulation of miR-26a-5p is likewise present during drug-induced liver injury, and when miR-26a-5p is instead upregulated under these conditions, it mediates the injury to the liver by targeting Bid [105], a protein involved in a pro-apoptotic pathway [106]. This caused a significant decrease in proinflammatory cytokines and a significant increase in anti-inflammatory cytokines, but more research must be completed to determine if the MSC-derived EVs directly protected the liver against acute liver injury during sepsis [89]. Another group of researchers used EVs from the same source to treat non-sepsis-induced ALI/ARDS in mice; when loaded with fluorescence, the mice showed increased fluorescence in the lungs up to 28 days after the initial treatment with the EVs [107]. When mice infected with *P. aeruginosa*, a highly virulent pathogen often leading to mortality in patients with cystic fibrosis or other underlying lung issues [108], were treated with the same type of EV–MSC-derived EVs, the mortality rate decreased from 80% down to 20% just 96 h after treatment. The clinical trial continued on healthy human patients to determine if there were any adverse reactions and found no changes in the patients’ vital signs or any adverse reactions or allergies. Furthermore, the nebulization of the EVs allowed them to avoid accumulation in the liver and spleen which often occurs when un-targeted EVs are injected [107]. While engineered EVs have yet to be extensively studied in clinical trials, the safety and efficacy of non-engineered EVs should inspire hope and excitement for the future of medicine. 

## 6. Current Challenges for Developing Engineered EVs

Thankfully, current research on the translational applications of engineered EVs is quite promising, but there are still challenges to overcome, most of which come from a logistical standpoint. Research has suggested that one of the most efficient methods of storing EVs long term, which was studied for up to two years in the referenced study, is at -80°C in a phosphate buffer solution with additional human albumin and trehalose as it allows for relatively easy EV recovery and does not impact the biodistribution of the EVs in mouse models [109]. However, this storage option is not likely feasible nor cost-effective for large-scale transportation from a laboratory to hospitals or clinical settings for the administration of EVs to patients. Furthermore, the mass production of EVs necessary for large-scale clinical applications proves difficult. The isolation of EVs is time-consuming, with the quicker methods being the most likely to damage EVs and produce low yields. Many approaches are also high cost. However, as most EVs, except for those derived from dendritic cells [6], do not contain MHC I or II proteins, they are unlikely to elicit an immune response if isolated from a donor rather than from a patient themselves [3,9,12,13]. More research must be performed to determine ways EVs can be isolated quickly, cost-effectively, and in a high-yield fashion before EVs can have widespread clinical applications. 

## 7. Further Research and Perspectives

In addition to the research necessary to allow a large-scale use of EVs translationally from a logistical standpoint, clinical trials on engineered EVs are greatly needed. The beginning stages of clinical trials using non-engineered EVs have begun and show promising results, but engineered EVs hold the most potential for efficacious treatment of diseases, including the ever-elusive highly effective cancer treatment. Current available targeting units used for generating engineered EVs have been summarized in Table 4. 

Moreover, this is especially true when considering their potential when used as a hybrid delivery system. That is when EVs are fused with liposomes to alleviate some of the existing drawbacks of EVs and liposomes alone. Liposomes have a long retention time in the blood when they are PEGylated and research is currently beginning on both how to fuse EVs and liposomes, as well as how the hybrid delivery vehicle may be better and more efficacious than either one individually [110,111]. Interestingly, one study found that using the hybrid EV/liposome increased the effectiveness of the delivery to breast cancer cells compared to both EVs and liposomes alone [112]. Another area of interest is the packaging of nanoreactors, which treat cancer in a variety of ways, inside of EVs. These nanoreactors have a variety of components that allow them to provide chemotherapeutic reagents, photothermal therapy sensitization, starvation therapy, and the generation of reactive oxygen species. When delivered in an EV without laser irradiation, they successfully destroyed 80% of cancer cells inside a 3D cancer cell sphere, and with laser irradiation, the cytotoxicity of cancer cells increased to 90% [113]. Future research should look to adapt these EV nanoreactors to be used in mouse models.

It has also been proposed that artificial intelligence (AI) could be used alongside EVs to generate more effective targeting peptides that easily perform click-chemistry and to analyze large amounts of data about a patient to best personalize their EV treatment [114]. For example, AI could look at a variety of biomarkers for a patient with breast cancer, and then determine that the most effective EV for their treatment would express a HELA targeting peptide and carry both a chemotherapeutic drug and a miRNA. More research must be completed in this area to elucidate the best practices for preparing, engineering, and administering these hybrids or nanoreactors EVs for translational applications, and perhaps AI may play a role in the future of EV research through the development of targeting peptides and the rapid personalization of EVs to the patient’s specific needs.

## Figures and Tables

**Table 1 ijms-24-15206-t001:** The applications of engineered EVs in cancer therapy.

Type of Cancer	Origin of EVs	Engineered strategy	Functional Cargo	Mechanism	Ref.
GBM	HEK293 T; human neural progenitor cells	Coincubation; click-chemistry	miR-124; anti-PD-L1 siRNA	Decreased expression of mcl-1 and c-Myc; decreased expression of PD-L1	[13,14]
Ovarian Cancer	HEK293 T; fibroblasts; M1 macrophages	Transfection; electroporation; extrusion approach	miR-92b-3p; miR-199a-3p; efferocytosis inhibitor MRX-2843	Suppressed expression of GABRA3; cell cycle arrest, increased expression of c-Met and mTOR; suppression of efferocytosis	[33,36,38,39]
Breast Cancer	Dendritic cells; MDA-MB-231; HEK293 T	Electroporation; transfection	Human neutrophil elastase (ELANE) and Hiltonol (TLR3 agonist); miR-205; miRNA let-7a; Dox	Antitumor immunity; chemotherapy; targeting of Bcl-2; downregulation of c-Myc	[41,42,43]
Cervical Cancer	Bone marrow-derived MSC	Transfection	miR-375	Ferroptosis	[48]
Melanoma	Melanoma cell lines	Transfection	miR-195-5p	Induced hypodiploidy	[56]
Gastric Cancer	HEK293 T	Transfection	circDIDO1	Increased expression of SCOS2	[57]
Colon Cancer	HEK293 T	Transfection; electroporation	miR-21 inhibitor; 5-FU	Cell cycle arrest; chemotherapy	[61]
Pancreatic Cancer	PANC-1; U937	Coincubation; sonication	Paclitaxel	Chemotherapy	[65]
Liver Cancer	HEK293 T	Transfection; sonication	miR-26a; CD47; Erastin; Rose Bengal	Targeting of ARPP19 and CDC6; ferroptosis; photosensitization	[50,67,68,69,70,71]
Osteosarcoma	Osteosarcoma cells	Transfection	lncRNA MEG3	Sponging of miR-185-5p	[73,74]

**Table 3 ijms-24-15206-t003:** Summary of engineered EVs in neurological diseases.

Type of Neurological Disease	Origin of EV	Engineered Strategy	Functional Cargo	Mechanism	Ref.
Alzheimer’s Disease	Dendritic cells	Electroporation; transfection	BACE1 siRNA	Inhibition of BACE1 protease	[12,91]
Parkinson’s Disease	Dendritic cells	Transfection	shRNA	Decreased expression of alpha-synuclein	[83]
Cocaine-induced Neuroinflammation	Dendritic cells	Transfection	miR-124	Halting activation of microglial cells	[84]
Stroke; Cerebral Ischemia	Brain endothelial cells	Coincubation	Mitochondria; HSP27	Decrease tight junction permeability; protect brain endothelium	[85]

**Table 4 ijms-24-15206-t004:** Summary of targeting units used for engineered EVs.

Target	Composition of the Unit	Engineered Strategy (Method of Addition)	Application Mentioned	Ref.
αvβ3 integrin	RGD; cRGD	Click-chemistry	Various cancer cell types within tumor and vasculature	[24,25,26]
TNBC Cells	α-Lactalbumin	Transfection	TNBC	[41]
EGFR	GE11 peptide (YHWYGYTPQNVI)	Transfection	Breast cancer cells overexpressing EGFR	[42]
Her2 protein	target-Her2-LAMP2-GFP	Transfection	Colon cancer cells overexpressing Her2	[61]
Apo-A1	SR-B1 receptor	Transfection	Liver cancer cells overexpressing Apo-A1	[67]
PD-L1	Anti-PD-L1	Transfection	Tumor cells expressing PD-L1	[77]
Alpha-B crystalline	CSTSMLKAC	Conjugation reaction	Infarcted heart tissue	[80]
Acetylcholine receptor	RVG peptide (YTIWMPENPRPGTPCDIFTNSRGKRASNG)	Transfection	Neurons	[12]
αvβ3 integrin	RGD-C1C2 (RGD fused to phosphatidylserine-binding domains of MFGE8)	Click-chemistry	Ischemic brain tissue	[17]

## Data Availability

Not applicable.

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
