# Peer review of "Engineered Extracellular Vesicles: Emerging Therapeutic Strategies for Translational Applications"

_ijms, 2023, doi:10.3390/ijms242015206_

Round 1

Reviewer 1 Report

The topic of creating and using extracellular vesicles is quite popular in the modern scientific world. Also, the use of extracellular vesicles is one of the promising areas of personalized medicine, along with hemosorption.

The authors identify 3 main directions in the design of EVs: anti-cancer, cardiological, neurological, and others (in the part of Engineered EVs in other human diseases). From the point of view of practical application, everything is described in great detail. But this is a wish primarily for further work.

However, since the review is devoted to the construction of extracellular vesicles, it sorely lacks a methodological part on methods for assessing the resulting vesicles. I understand that this way any review can become infinitely large and overloaded. But at the end of each section, it would be desirable to list the main approaches to obtaining, isolating and characterizing the resulting extracellular vesicles.

From the point of view of literature analysis, the review shows not bad results. Of the 111 references, only 33 refer to publications older than five years. Of these, no more than 10 are older than 10 years, which it is impossible not to mention when introducing readers to the topic of the review.

Author Response

Responses to Reviewers:

We thank the editor for this opportunity to revise our manuscript, and appreciate all the constructive suggestions and comments from the reviewers. Changes to the revised manuscript are marked in red text. To further address and clarify the concerns of reviewers, we would like to provide a point-by-point response as follows:

 Reviewer #1

The topic of creating and using extracellular vesicles is quite popular in the modern scientific world. Also, the use of extracellular vesicles is one of the promising areas of personalized medicine, along with hemosorption.

The authors identify 3 main directions in the design of EVs: anti-cancer, cardiological, neurological, and others (in the part of Engineered EVs in other human diseases). From the point of view of practical application, everything is described in great detail. But this is a wish primarily for further work.

Response: Thank you for your positive comments.

However, since the review is devoted to the construction of extracellular vesicles, it sorely lacks a methodological part on methods for assessing the resulting vesicles. I understand that this way any review can become infinitely large and overloaded. But at the end of each section, it would be desirable to list the main approaches to obtaining, isolating and characterizing the resulting extracellular vesicles.”

Response: Thank you for your constructive suggestions and comments. We added one table (Table 2) to summarize the main approaches for obtaining, isolating and characterizing the resulting extracellular vesicles.

From the point of view of literature analysis, the review shows not bad results. Of the 111 references, only 33 refer to publications older than five years. Of these, no more than 10 are older than 10 years, which it is impossible not to mention when introducing readers to the topic of the review.

Response: Thanks for your suggestion, we highlighted this point in abstract part.

Reviewer 2 Report

hello

thank you for an interesting review

recent improvements in molecular and genetic studies are essential for nowadays medicine

please highlight the future role of the extracellular vesicles and their impact on medicine

add the aim of this review in the abstract and introduction section

please highlight more whats new in the topic

thank you 

Author Response

Responses to Reviewers:

We thank the editor for this opportunity to revise our manuscript, and appreciate all the constructive suggestions and comments from the reviewers. Changes to the revised manuscript are marked in red text. To further address and clarify the concerns of reviewers, we would like to provide a point-by-point response as follows:

Reviewer #2

“Thank you for an interesting review, recent improvements in molecular and genetic studies are essential for nowadays medicine”

Response: Thank you for your positive comments.

“please highlight the future role of the extracellular vesicles and their impact on medicine, add the aim of this review in the abstract and introduction section; please highlight more what’s new in the topic”

Response: Thank you for your constructive suggestions and comments. We have highlighted the future role of EVs and their impact on medicine in the introduction section from page 7 to page 8 (highlighted in red in text), added the aim of this review in the abstract and introduction section, and highlighted advantages and therapeutic promises of engineered EVs in abstract (highlighted in red in text).
